# Effectiveness of a Primary Care Multidisciplinary Treatment for Patients with Chronic Pain Compared with Treatment as Usual

**DOI:** 10.3390/jcm12030885

**Published:** 2023-01-22

**Authors:** Rinske M. Bults, Johanna M. van Dongen, Raymond W. J. G. Ostelo, Jo Nijs, Doeke Keizer, C. Paul van Wilgen

**Affiliations:** 1Department of Rehabilitation Science, Faculty of Physical Education and Physiotherapy, Vrije Universiteit Brussel, 1050 Brussels, Belgium; 2Pain in Motion International Research Group, Department of Physiotherapy, Human Physiology and Anatomy, Faculty of Physical Education & Physiotherapy, Vrije Universiteit Brussel, 1050 Brussels, Belgium; 3Department of Health Sciences, Faculty of Science, Vrije Universiteit Amsterdam, Amsterdam Movement Sciences Research Institute, 1105 Amsterdam, The Netherlands; 4Department of Epidemiology and Data Science (Amsterdam UMC, Location VUmc), Amsterdam Movement Sciences Research Institute, 1046 Amsterdam, The Netherlands; 5Chronic Pain Rehabilitation, Department of Physical Medicine and Physiotherapy, University Hospital Brussels, 1050 Brussels, Belgium; 6General Practice “Het Homeer”, 9281 Harkema, The Netherlands; 7Transcare, Transdisciplinary Pain Management Center, 9711 Groningen, The Netherlands

**Keywords:** chronic pain, primary care, multidisciplinary, treatment, effectiveness

## Abstract

To manage chronic pain, multidisciplinary interventions have been increasingly deployed, mostly in secondary or tertiary care settings. Evidence on the effectiveness of multidisciplinary intervention within primary care is scarce. This study examined the effectiveness of a primary care multidisciplinary treatment for chronic pain compared with treatment as usual (TAU). The intervention consisted of pain neuroscience education and treatment by a GP, psychologist, and physiotherapist. Both groups filled out patient-reported outcome measures at baseline, 6 months, and 12 months. The results indicated there were no statistically significant differences for the primary outcomes of pain intensity, number of pain sites, and health-related quality of life (HR-QoL). There was a statistically significant difference in the secondary outcome perceived health change in favor of the intervention group. None of the other differences were statistically significant. A post-hoc analysis showed that there were statistically significant effects on patients’ illness perceptions in favor of the intervention group. Based on the results, the findings do not support effectiveness of a low intensity outpatient multidisciplinary primary care treatment to treat chronic pain compared with TAU. However, as a result of several study limitations, it is considered unwarranted to conclude that multidisciplinary treatment in primary care is not valuable at all.

## 1. Introduction

Chronic pain is a significant problem for a large number of people. It is estimated that about 20% of the European population suffers from chronic pain [1]. The majority of patients with chronic pain are treated within primary care, with less than 2% of patients visiting a pain clinic [1]. Despite guidelines advising multidisciplinary care for patients with chronic pain, most patients receive monodisciplinary care and/or ‘typical’ biomedical interventions, such as medication (i.e., opioids) or surgical interventions [1]. There has been growing concern over the use of biomedical interventions, as evidence of effectiveness is lacking and these interventions may instead cause harm [2].

Multi- or transdisciplinary assessments and treatments are recommended because chronic pain is a biopsychosocial problem, in which pain is influenced and maintained by physical, psychological, and social factors [3]. Negative illness perceptions, anxiety, and depression as well as social factors are associated with chronic pain, and might worsen pain and pain-related disability [4,5]. With the biopsychosocial model in mind, multidisciplinary treatments for chronic pain have increasingly been deployed in the last few decades [6,7,8]. Multidisciplinary treatments often combine cognitive behavioral therapy, pain education, physical approaches (e.g., graded activity), exposure therapy, and/or relaxation techniques [9]. Such programs usually focus on pain management and improved functioning, rather than cure or pain relief. Nevertheless, they have shown to be successful in decreasing symptoms and increasing functioning [9,10]. Although most research on multidisciplinary treatment for chronic pain is conducted in secondary or tertiary care settings, there is evidence that multidisciplinary treatment in primary care is effective. That is, primary care multidisciplinary treatment appears to lead to lower healthcare utilization, reduced sick leave, improved depression, increased social activity, reduced medication use, and reduced disability [11,12,13].

Given the above, current guidelines for the treatment of chronic pain recommend a multidisciplinary approach within the primary-care setting [2,14,15]. While it is widely acknowledged that the goal is to strive towards establishing a multidisciplinary team for managing chronic pain, much of the responsibility for managing patients with chronic pain still falls upon primary care providers [16]. However, many primary care providers (often general practitioners) have neither the time, nor the pain management training to effectively treat chronic pain [17]. Additionally, it is not sufficient to simply create a team from different healthcare disciplines: the treatment must be fully integrated across these disciplines to achieve the best results [16].

There are several potential barriers that obstruct the implementation of such an approach, such as the readiness of different health disciplines, structural supports from the health system, scheduling issues, lack of physical space, and the ability to bundle costs [16]. As a result of these barriers, multidisciplinary treatment in primary care has not yet been widely implemented. Furthermore, there is a lack of high-quality evidence on the effectiveness of multidisciplinary treatment in primary care. Therefore, this study aimed to evaluate the effectiveness of a multidisciplinary treatment program embedded in primary care compared with treatment as usual (TAU) for patients with chronic pain.

## 2. Materials and Methods

### 2.1. Design

A multicenter, non-randomized, controlled trial was conducted with a 12-month follow-up period.

Two groups of patients were included in this study: an intervention group and a control group. The study design and informed consent procedure were approved by the Medical Ethics Committee of Amsterdam UMC, location VUMC. The trial is registered under number NTR6014 (https://trialsearch.who.int/Trial2.aspx?TrialID=NTR6014, accessed on 16 January 2023).

### 2.2. Sample Size

From an a priori power calculation (power = 0.80, α = 0.05), we concluded that 28 participants should be included per group to detect a clinically relevant between-group difference of 2 points on the VAS (SD = 2.7). Considering a dropout rate of 30%, it was concluded that a total of 80 participants should be included in this study (40 participants per group). The 2-point difference on the VAS was based on the previously identified Minimal Important Change of the NRS (or VAS) of 2 points [18]. In addition, two studies that specifically explored which difference is considered worthwhile by patients, if no treatment is compared with conservative treatment for low back pain, concluded that a difference between these groups of approximately 20% is considered worthwhile by patients [19,20]. The SD is based on a study in Denmark which compared pain intensity after a multidisciplinary treatment with pain intensity after general practice [21].

### 2.3. Participants

Both study groups consisted of patients with chronic non-specific pain. To be eligible for the current study, patients had to experience chronic non-specific pain for a period longer than 6 months, had to be aged between 18–70 years, had to have consulted a general practitioner for their chronic pain during the last 3 months, and had to have no prior treatment with cognitive behavioral therapy aimed at treating chronic pain-related psychological issues. Exclusion criteria were: pain as a result of cancer or neurological conditions (e.g., diabetes, migraine), being diagnosed with cognitive impairments, current severe psychiatric problems, not able to complete online questionnaires, or not able to speak/understand Dutch.

Participants for the intervention group were recruited at three general practices. General practitioners (GPs) working at these practices identified eligible patients in their practice, and invited them to take part in the multidisciplinary program. The multi/transdisciplinary program under study was implemented in these practices several years before the trial started. All patients who entered the multidisciplinary program in these practices and met the in- and exclusion criteria were asked to participate in the trial. All patients were provided with an informed consent form, which they were asked to sign upon entering the trial.

Participants for the control group were recruited via advertisements on social media (i.e., Facebook). Control group patients did not receive the treatment under study; instead, they received treatment at general practices other than those providing the intervention. To keep both groups as similar as possible, only patients living in the same geographical area as the patients in the intervention group (i.e., the North of the Netherlands) were recruited for the control group. Upon entering the study, all participants received written information about the study and were to provide informed consent.

### 2.4. Data Management

Data were handled in accordance with the Dutch Personal Data Protection Act. Data were gathered electronically using survey software (NetQuestionnaire, now Survalyzer www.survalyzer.com). All data were anonymized by removing personal information. Data were exported to SPSS and stored on a password-protected computer. The password was changed on a regular basis. Participant files were stored for a period of 5 years after the completion of the study, after which the files were removed.

### 2.5. Intervention Program

Participants in the intervention group received a multidisciplinary treatment provided by a psychologist, a physiotherapist, and a general practitioner trained in pain management. The training of the multidisciplinary team consisted of a two-day course about performing a multidisciplinary assessment, pain neuroscience education, and multidisciplinary pain management interventions. In the implementation phase, the team received an additional two-day training on the job by an expert in pain management interventions.

The program was set up as follows:

#### 2.5.1. Diagnostic Process

a.Pre-intake: After being referred for a multidisciplinary assessment, patients were asked to fill out patient-reported outcome measures (PROMS), assessing various outcomes, such as pain-related outcome measures and socio-demographic characteristics (described below in Section 2.6). PROMS were used to determine the predominant pain mechanism (i.e., central sensitization, neuropathic pain, or nociceptive pain) and adapt the intervention to the patients’ specific symptoms and needs. In this phase, specific attention was paid to patients showing the following symptoms of central sensitization:
Pain NRS > 6Widespread Pain Index (WPI) ≥ 7Central Sensitization Inventory (CSI) > 40Pain Catastrophizing Scale (PCS) > 30b.Assessment: a 3 h multidisciplinary assessment according to a matched care model conducted by a physician, psychologist, and physiotherapist. For the assessment, a common assessment model was used in which pain symptoms, behavioral, somatic–medical, emotional, social, and cognitive factors, and the stages of change were determined. This model is based on the PSCEBSM model in which an appraisal is made of patients’ type of pain: (P), somatic and medical factors (S), cognitions and perceptions (C), emotional factors (E), behavioral factors (B), social factors (S), and motivation (M) [22]. Both the GP and the physiotherapist conducted a physical examination (both ±30 min in duration) to assess the patient’s physical status (i.e., determine pain type, mobility, and pain behavior). After the assessment, the multidisciplinary team discussed the patient’s pain type and constructed a biopsychosocial model of the patient, their pain, and the factors contributing to the current symptoms. This model served as a framework for pain education and further treatment.

#### 2.5.2. Treatment Process

Depending on the information found in the assessment and the biopsychosocial factors that were determined, a treatment plan was made through shared decision making. Every patient received pain neuroscience education about their pain [23,24], mostly using the sensitization model [25], in which patients were educated about their pain and the factors that influence their pain.

a.Pain Neuroscience Education (PNE): This education consisted of two one-hour appointments, following the guidelines by Nijs et al. [26]. The first session was conducted by the general practitioner and during this session the medical and somatic aspects of the patients’ pain were discussed, and—in case of central sensitization (i.e., “Increased responsiveness of nociceptive neurons in the central nervous system to their normal or subthreshold afferent input” as defined by the IASP [27]), this phenomenon was explained [28]. During the second session, the psychologist and physiotherapist started by discussing the most essential information provided by the general practitioner and repeated the principles of central sensitization if applicable. The main goal during this session was to educate patients about the biopsychosocial factors related to/maintaining their altered pain sensitivity i.e., central sensitization. In addition, through shared decision making, a treatment plan was discussed in which these factors were addressed.b.Further treatment: Further possible treatment depended on the presence of patient-specific factors:

Depression, anxiety, fear of movement → psychological treatment such as cognitive behavioral treatment, acceptance and commitment therapy, relaxation techniques, and/or exposure therapy. In case of more severe anxiety and/or depressive symptoms, medication such as antidepressants could be (temporarily) prescribed by the general practitioner.Post-traumatic stress disorder (PTSD) → psychological treatment such as trauma therapy (e.g., EMDR or imaginary exposure).Lack of exercise, avoidance behavior, and/or disuse → physical therapy such as graded activity, pacing programs and/or graded exposure, encouragement of healthy exercise habits (e.g., 30 min walk/cycle every day, daily activity program)Persistence behavior → daily schedule with alternating periods of activity and rest/relaxation. Helping patient decide where to set boundaries, usually under guidance of psychologist and physiotherapist.Use of medication such as opioids → tapering off medication by the general practitioner or switching to less harmful medication (such as antidepressants).

Generally, patients saw the psychologist for a total of 5–12 sessions (on average 45 min in duration) and the physiotherapist for 9–20 sessions (on average 25 min in duration). The general practitioner met with the patient during initial assessment for the Pain Neuroscience Education session and then when necessary (e.g., for tapering off medication, prescribing medication). More information about this treatment can be found at www.transcare.nl (accessed on 16 January 2023, in Dutch).

Control group participants received TAU for chronic pain. Dutch general practitioners typically work according to the “NHG-Standaard Pijn”, a guideline for general practitioners in managing pain [14]. The key elements of the guideline are as follows:Education about chronic pain and the biopsychosocial modelNon-drug treatment: advise to seek distraction and support and to find a good balance between relaxation and activity. Possible referral to a psychologist in case of harmful cognitions, emotions, and behaviors; possible referral to a physiotherapist or remedial therapist for exercise programs aimed at an active lifestyle; and possible referral to social worker if social problems play a role.Drug treatment: avoid drug treatment if possible, especially avoid opioids if possible. When prescribing drugs, aim to prescribe for a short period.Referrals: referral to a rehabilitation if patient is experiencing a high degree of limitations due to pain; referral to medical specialist is there is a possible underlying cause that is treatable, etc.

### 2.6. Outcome Measures

Baseline measures included demographic variables such as sex, age, civil status, education level, and work status. Effect measures were assessed at baseline, and at 6- and 12-month follow-up points.

### 2.7. Primary Outcome Measures

Pain intensity was determined using an 11-point Numeric Rating Scale (NRS), ranging from 1 (no pain) to 10 (severe pain) [29].

The number of pain sites was measured by the Widespread Pain Index (WPI), on which participants can indicate where the pain is located using an illustration of a human body [30]. The number of pain sites is used as an overall score, ranging from 1 to 19.

Health-related Quality of Life (HR-QoL) was measured using the RAND-36, a self-administered 36-item questionnaire. It consists of 8 subscales, i.e., physical functioning, social functioning, role limitations due to physical problems, role limitations due to emotional problems, mental health, vitality, pain, and general health perception [31]. In addition, one item measures the experienced health change over the past year. For this study, the Dutch version was used, which is shown to be a reliable instrument for measuring HR-QoL [32]. The participants’ RAND-36 responses were converted to utility values by applying the scoring algorithm developed by Brazier et al. [33]. Utility values indicate the preference of individuals’ for a certain health state and are anchored at 0.0 (equal to death) and 1.0 (equal to full health).

### 2.8. Secondary Outcome Measures

Symptoms of central sensitization was measured using the Central Sensitization Inventory (CSI) [34,35]. The CSI is a screening measurement to identify symptoms of central sensitization (CS). CS is a physiological phenomenon in which the central nervous system becomes hypersensitive to stimuli [28]. For this study, only Part A was analyzed, which assesses 25 health-related symptoms that are common to CS. Total scores range from 0–100. The CSI was found to be psychometrically sound, with good test–retest reliability and internal consistency [34]. Studies found that a score of 40 or higher distinguishes between patients with CS and patients without CS [36].

Pain catastrophizing was measured using the Pain Catastrophizing Scale (PCS) [37], a 13-item scale that assesses catastrophizing in clinical and nonclinical populations. Each item is rated on a 5-point scale ranging from 0 (not at all) to 4 (all the time). The PCS was found to be a valid and reliable instrument in measuring pain catastrophizing [37,38].

Patients’ satisfaction with the received treatment was measured once, at 12 months, using a shortened version of the CQ-Index Module Pain [39]. The questionnaire consisted of 4 items. Patients were asked to rate on a Likert scale how seriously they felt their pain was taken by the healthcare professionals ranging from 0 (never) to 3 (always), whether they trusted the healthcare professionals’ expertise from 0 (never) to 3 (always), how they rated the result of the treatment from 0 (bad) to 4 (excellent), and how they rated the care they received from the healthcare professionals from 0 (very bad) to 10 (excellent).

### 2.9. Post Hoc Analysis

Two outcome measures, which were not described in the original study protocol as registered in the trial register, were added in a post hoc analysis.

Illness perceptions were measured using the Brief Illness Perceptions Questionnaire (IPQ-B [40]). This instrument measures individuals’ perceptions of illness with 9 items. Five of the items assess cognitive illness representations: consequences, timeline, personal control, treatment control, and identity. Two items assess emotional representations: concern and emotional response. One item assesses illness comprehensibility: understanding. The last item is an open-ended response item on which patients are asked to list the three most important causes for their illness. The IPQ-B shows good test–retest reliability and concurrent validity and demonstrates good predictive and discriminant validity [41]. For this study, only the first 8 items were analyzed. The last item about causal appraisal was not used in this study.

Anxiety and depression were measured using the Hospital Anxiety and Depression Scale (HADS) [42]. This instrument identifies cases of anxiety and depression among nonpsychiatric hospital clinics. It is divided into an anxiety subscale and a depression subscale, which both contain seven items. The HADS was found to be able to assess symptom severity and caseness of anxiety and depression in both somatic, psychiatric, and primary care patients and in the general population [43].

### 2.10. Analytical Methods

For all analyses, a two-tailed significance level of *p* < 0.05 was considered statistically significant.

Descriptive statistics were used to describe baseline characteristics of both groups.

The primary analysis was an intention-to-treat analysis. All outcomes were analyzed using a linear mixed model fitted by maximum likelihood, with responses at baseline, 6 months, and 12 months. The model had a 2-level structure, with repeated measures (level 1), nested within patients (level 2). The effect of interest was the overall difference in outcomes between groups during the complete duration of follow-up. Additionally, time by treatment interactions were added to the models to assess the differences in outcomes per time point. Regression coefficients with 95% confidence intervals (CIs) were estimated. To deal with the non-randomized nature of this study, effect differences were adjusted for patients’ propensity score. The propensity score indicates the probability of a patient being assigned to the intervention group, given a set of baseline characteristics. The propensity score was estimated based on the following possible confounding variables: whether patients had other complaints besides pain, whether patients had their pain symptoms for the first time or not, the duration of their pain at baseline, employment status, education level, age, and sex using the pscore package in STATA [44]. As for clinical relevance, a 30% improvement rate within group difference was deemed a clinically meaningful improvement [45].

Differences between satisfaction rating were analyzed using independent t-tests.

A post hoc analysis was performed on the IPQ-B and the HADS to gain more insight in how illness perceptions, anxiety, and depression changed over time. This was investigated by a linear mixed model fitted by maximum likelihood as well.

All analyses were performed in SPSS v25 and STATA v15.

## 3. Results

### 3.1. Participants

In total, 107 participants were recruited for the study. Of them, 89 filled out the baseline questionnaire (43 in the intervention group; 46 in the control group) and were included in the study (Figure 1).

### 3.2. Participants’ Characteristics

In Table 1, the participants of both groups are described. Baseline characteristics were mostly comparable across groups. In both groups, the large majority were female, married, had children, and most of them had a middle level of education. However, in the control group, most participants did not have paid employment, while the majority of the participants in the intervention group were employed. Only a small proportion of both groups were smokers.

### 3.3. Intention-to-Treat Analysis

The mean differences between groups for the primary outcome measures are shown in Table 2. The values presented under average between group differences are model estimates of linear mixed-effect models and are corrected by baseline characteristics and propensity scores. The regression coefficients can be interpreted as average differences between the intervention group and the control group at a certain follow-up point (6 or 12 months) compared with the baseline. The overall effect measures can be interpreted as the effect over the total follow-up time of 12 months, instead of time x treatment effects.

The overall differences between both groups for pain intensity (mean difference = −0.9, 95% CI: (−1.9, 0.05)), number of pain sites (mean difference = −1.2, 95% CI: (−2.5, 0.1)) and HR-QoL (mean difference = 0.02, 95% CI: (−0.02, 0.05) were not statistically significant. The average between-group difference on pain intensity (0.9-point difference) was also not clinically relevant (<2 points difference).

While the overall effect (i.e., the difference in effects over the whole 12 months) was not statistically significant, there was a statistically significant difference between both groups in the number of pain sites at 12 months (i.e., the difference in effect between 6 and 12 months). In addition, there was a statistically significant difference between both groups in HR-QoL at 12 months. None of the within-group improvements in the intervention group were clinically relevant either (with a 14% improvement in pain intensity, a 6% improvement in number of pain sites, and an 8% improvement in HR-QoL). Likewise, the within-group improvements in the control group were not clinically relevant either (a 4% improvement in pain intensity, an 8% improvement in number of pain sites, and a 10% improvement in HR-QoL).

The mean differences between groups for the secondary outcome measures can be found in Table 3. At 6 months, the intervention group rated their overall health (as measured on the RAND-36 General Health item) statistically significantly better than the control group, but the corresponding overall effect (over the whole 12 months) was not statistically significant. Additionally, there was a statistically significant overall effect on perceived health change (RAND-36 Health Change), where the intervention group indicated their health as being improved more compared with the control group. In addition, the within-group improvement in the intervention group was clinically relevant (30% improvement). All other effects were not statistically significant. As for clinical relevance, there was a clinically relevant within-group improvement in role functioning due to the physical problems subscale in the intervention group (45% improvement), while there was no clinically relevant within-group improvement in the control group. All other changes were deemed not clinically relevant (<30% improvement).

### 3.4. Patients’ Satisfaction

As can be seen in Table 4, patients in the intervention group rated their treatment more positively on all items. However, only the difference in how much they felt they were taken seriously by the healthcare professionals was statistically significant.

### 3.5. Post Hoc Analysis: IPQ-B

In Table 5, the results of the post hoc analyses can be found. In the control group, scores on the IPQ-B items tended to remain unchanged or increase, while in the intervention group, scores tended to decrease over time. There were statistically significant overall effects in favor of the intervention group on the items of consequences, timeline, concern, and emotional response. The overall effects on the other items were not statistically significant. There was also a statistically significant difference between the intervention group and control group at 12 months on the item of identity, but the corresponding overall effect was not statistically significant. None of the within-group differences in illness perception, anxiety, or depression were deemed clinically relevant (i.e., >30% improvement).

## 4. Discussion

This study evaluated the effectiveness of a multidisciplinary treatment for chronic pain embedded in primary care and compared with treatment as usual. The results show that there were no statistically significant effects, nor clinically relevant between-group or within-group differences, on the primary outcomes pain intensity, number of pain sites, and HR-QoL. As for the secondary outcomes, there was a statistically significant difference in Health Change (measured on the RAND-36) in favor of the intervention group. There were no statistically significant overall effects on the other secondary outcomes: symptoms of central sensitization, pain catastrophizing, depression, anxiety, physical functioning, role functioning due to physical problems, energy, emotional wellbeing, role functioning due to emotional problems, social functioning, and general health.

In the post hoc analysis, there was a statistically significant effect on illness perceptions, with the intervention being significantly more effective in decreasing negative illness perceptions and increasing perceived health after 12 months compared with TAU. However, the changes in illness perceptions in the intervention group were not deemed clinically relevant.

The results show that there is no statistically significant nor clinically relevant effect on pain intensity. A possible explanation for the lack of a statistically significant effect on pain intensity and other pain-related outcomes is the duration of pain for the participants in this study. In the current study, both the participants in the intervention group and in the control group had had their pain for a substantial amount of time prior to the intervention (103 months in the control group and 93 months in the intervention group). Chronicity of pain has been shown to be associated with only modest changes in pain severity, self-reported disability, pain catastrophizing, and fear of movement/re-injury [46]. This seems to underline the need for an earlier multidisciplinary approach in patients with chronic pain, as not recognizing psychosocial risk factors can lead to chronification of pain [47]. Moreover, it is possible that the intervention was not intensive enough for the patients in this study. Research has shown that highly intensive multidisciplinary inpatient programs are more effective than low-intensive outpatient programs in managing chronic pain [9]. This intervention can be classified as a relatively low-intensive outpatient treatment, which may not have been sufficiently intensive enough to have a statistically significant effect on pain-related outcomes. In addition, it may be worthwhile investigating the effect of offering more or longer pain neuroscience education sessions, as a recent systematic review found favorable effects of longer duration of PNE, although this needs further study [48]. In addition, at 12 months, the PNE-session had been 12 months in the past for patients, and the intervention (i.e., sessions with the physiotherapist and/or the psychologist) may not have been active for several months for many patients at this stage as well. Looking at the results, most of the changes in the intervention group as well as between-group differences appeared to occur in the first 6 months, when the patients were often still in active therapy. Further studies could focus on adapting treatment, such as adding booster sessions in which the pain neuroscience education is repeated after a certain amount of time, or offer more intensive treatment in an effort to improve the long-term effects of the intervention.

Furthermore, the primary health care providers in the intervention were trained to work with the protocol. Some of them were, however, biomedically trained doctors and physiotherapists. It is a possibility that the total amount of training for the primary health care providers was too short to get acquainted with the protocol. Further studies may investigate the effect of more extensive training of the multidisciplinary team providing the intervention, and in addition, check whether the team’s attitude and behavior towards chronic pain did indeed change after training. In addition, the pre-existing relationship (i.e., prior to starting the intervention) between the patient and the care provider might have been a barrier for changing from a biomedical to a biopsychosocial approach.

Previous research has found that the belief that one has control over pain is strongly associated with decreases in pain intensity and patient-rated disability [49]. In this study, patients’ beliefs about personal and treatment control were moderate at best and did not statistically significantly increase during the course of the study, which could further explain the lack of effect on pain intensity and disability. Lastly, patients in both the control group and the intervention group reported significant levels of depression (HADS > 8) at baseline, and symptoms of depression appeared to increase over time. Research has shown that depression is a predictor of poor outcome for multidisciplinary treatment and quality of life in patients with chronic pain [50,51]. While levels of depression were equal in both groups (and differences were not statistically significant), this could dampen the effect of treatment success. It might also be worth investigating the effect of depression on the effect of the intervention and focus on how specific types of pain respond to multidisciplinary treatment compared with TAU.

A post hoc analysis showed that in the intervention group, patients’ illness perceptions changed statistically significantly compared with patients in the control group. This is possibly the result of the pain neuroscience education sessions the intervention group received, which provided patients with an in-depth biopsychosocial explanation of their pain and addressed the often-incorrect somatic perceptions patients had about their pain. The intervention led to the result that patients felt that their pain affected their life less, that they think their pain will continue for a shorter duration, that they were less concerned about their pain, and that they were less affected by their pain emotionally. In addition, patients in the intervention group perceived their health to have improved significantly more the past year compared with patients in the control group. Thus, the intervention seemed to mostly affect patients’ perception of their situation, while not necessarily improving other outcomes.

It is possible that reporting on pain and pain-related outcomes itself had an effect on patients’ outcomes in both the intervention group and the control group. Research has shown that attention to pain and mood both alter pain perception, and filling out the self-reports may have drawn attention to the pain and pain-related outcomes and hence influenced the patients’ experience of their pain and its outcomes [52]. However, as both the control group and the intervention group received treatment (treatment as usual vs. intervention), the effect of self-reporting is believed to be equal in both groups and it is not deemed to have led to bias.

It was not surprising that there was no statistically significant difference in how patients rated the result of their treatment, as there was no significant difference in most pain-related outcomes. However, patients in the intervention group did feel they were taken more seriously by healthcare professionals compared with patients in the control group.

### 4.1. Comparison with Previous Studies

Previous studies found favorable results for multidisciplinary care for chronic pain patients, mostly in intensive inpatient settings. Similar results as the results found in the current study on patients’ beliefs were found by Moss-Morris et al. [53], which showed that after a 4-week multidisciplinary treatment for chronic pain, patients’ beliefs about consequences and emotional response statistically significantly decreased over time. In the study by Moss-Morris et al. [53], a decrease in patients’ negative illness beliefs was also accompanied by an improvement in pain-related disability and physical functioning, which was not found in the current study [53]. A possible explanation for this difference is the fact that the intervention in the aforementioned study was a highly intensive program in secondary care, which may result in better results. Research on multidisciplinary treatment in primary care was scarce, but the results seem similar to the results in the current study. A study by Åsenlöf and colleagues similarly found no significant differences between a tailored behavioral treatment and exercise-based treatment after a 2-year follow up on pain intensity, pain control, and functional self-efficacy [11]. A study by Westman and colleagues showed decreased health care utilization in the group receiving multidisciplinary treatment in primary care, but also did not show statistically significant differences on the SF-36, the PCS, and pain intensity compared with treatment as usual [13]. Stein and Miclescu found statistically significant effects of a primary care multidisciplinary rehabilitation treatment program on sick leave, health care utilization, social activity, and depression. Similar to the current study, however, there were no statistically significant effects on pain intensity and physical activity [12].

### 4.2. Strengths and Limitations

The most important strength of this study is that it is one of the first studies investigating multidisciplinary treatment for chronic pain embedded in primary care, as well as its patient-centered approach, the use of propensity scores to deal with its non-randomized design, and the relatively long follow-up period.

This study also has several limitations. First, the allocation of the participants across study groups was not randomized. Randomization was not possible at patient or cluster level because of the nature of the intervention and the lack of resources to randomize entire practices. Patients in both the intervention group and control group remained at the general practices that they were already registered at, making it impossible to perform an RCT or CRT. Therefore, convenience sampling was used. This could have caused a selection bias, which we tried to reduce by using propensity score weighting. Second, the sample size was relatively small, and the dropout of participants was considerable (i.e., 67.4% in the intervention group and 56.5% in the control group). However, the sample size was sufficient and as Twisk et al. [54] argued, longitudinal mixed models are well able to deal with missing values through their use of maximum likelihood estimation and result in valid estimates [55]. In addition, there was no differential dropout or attrition, as in the drop-out rates were similar in both arms (control group and intervention group) and, therefore, the risk of a biased result is considered low [55]. Third, it was not checked whether the participants in the control group were actually treated according to the guideline for managing patients with chronic pain; the “NHG Standaard Pijn” [14]. It is possible that healthcare professionals dealing with the control group deviated from the guideline. However, it is considered best practice and lawful that general practitioners follow guidelines as much as possible and deviate from it where necessary [56]. Deviations from the guideline can therefore be considered treatment as usual and do not influence the interpretation of the results. Likewise, while the healthcare professionals in the intervention group received training and were instructed to provide treatment according to protocol, it was not checked whether they actually adhered to protocol. The intensity of the intervention (i.e., the amount of therapy sessions a patient received from the physiotherapist and/or psychologist, the exact nature of the therapy sessions, how long the total duration of the intervention was for every patient, etc.) was not documented either and the effect of intervention intensity (i.e., dose-response relationship) could therefore not be taken into account. Future research is therefore recommended to include a fidelity check of the intervention. Furthermore, for future studies it is recommended to carefully document the type and frequency of therapy sessions within the intervention in order to be able to investigate dose-response relationships.

## 5. Conclusions

A multidisciplinary treatment for chronic pain embedded in primary care did not result in statistically significant nor clinically relevant improvement in pain intensity, number of pain sites, and health-related quality of life compared with treatment as usual. Secondary analyses showed that it had an effect on patients’ illness perceptions and perceived health, and that patients felt more taken seriously by their healthcare providers compared with patients who received treatment as usual. Based on these results, the findings do not support implementation of a low-intensity outpatient multidisciplinary primary care treatment to treat chronic pain. However, as a result of several study limitations, it is considered unwarranted to conclude that multidisciplinary treatment in primary care is not valuable at all. Future studies may aim to address the listed limitations to aim for a fairer, more detailed comparison between the intervention and TAU. Multidisciplinary treatment in primary care for chronic pain may very well be effective as well as cost-effective, but further research is necessary to investigate how the intervention has to be set up in order for it to be significantly more effective than TAU in managing chronic pain.

## Figures and Tables

**Figure 1 jcm-12-00885-f001:**
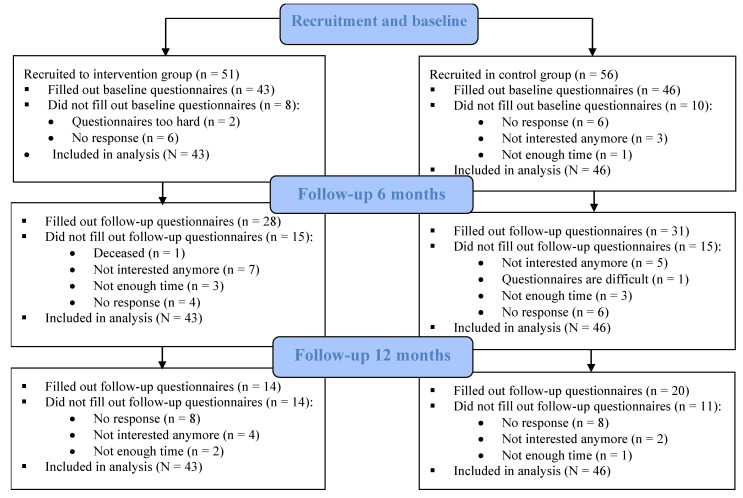
Flow chart of the participants.

**Table 1 jcm-12-00885-t001:** Demographic data of both groups.

	Control Group (*n* = 46)	Intervention Group (*n* = 43)
Sex: male *n* (%) ^1^	3.0 (6.5)	8.0 (18.6)
Age: mean (SD)	44.8 (14.3)	43.7 (13.9)
Marital status: *n* (%)		
Married	15.0 (32.6)	19.0 (44.2)
Divorced	4.0 (8.7)	3.0 (7.0)
Cohabiting unmarried	9.0 (19.6)	6.0 (14.0)
Single	9.0 (19.6)	7.0 (16.3)
Living with parents	1.0 (2.2)	2.0 (4.7)
Widowed	0.0 (0.0)	1.0 (2.3)
Unknown	8.0 (17.4)	5.0 (11.6)
Children yes: *n* (%) ^1^	22.0 (47.8)	25.0 (58.1)
Not answered	8.0 (17.4)	6.0 (14.0)
Education level: *n* (%)		
No education	0.0 (0.0)	1.0 (2.3)
Primary school	3.0 (6.5)	1.0 (2.3)
Lower education	8.0 (17.4)	8.0 (18.6)
Middle education	20.0 (43.5)	20.0 (46.5)
Higher education	7.0 (15.2)	6.0 (14)
Unknown	8.0 (17.4)	7.0 (16.3)
Paid work yes: *n* (%) ^1^	13.0 (28.3)	25.0 (58.1)
Unknown	3.0 (6.5)	5.0 (11.6)
Allowance: *n* (%)	19.0 (43.3)	12.0 (27.9)
Sickness benefit	0.0 (0.0)	6.0 (14.0)
Disability Benefits Act	1.0 (2.2)	1.0 (2.3)
WAJONG ^2^	6.0 (13)	1.0 (2.3)
Unemployment Law	5.0 (10.9)	2.0 (4.7)
Welfare	7.0 (15.2)	2.0 (4.7)
Unknown	8.0 (17.4)	8.0 (16.3)
Smoker yes: *n* (%) ^1^	9.0 (19.6)	10.0 (23.3)
Unknown	8.0 (17.4)	7.0 (16.3)
Pain duration in months: mean (SD)	103.0 (113.0)	98.0 (157.0)

^1^ Dichotomized; ^2^ Wet arbeidsongeschiktheidsvoorziening jonggehandicapten (Work and Employment Support for Disabled Young Persons Act).

**Table 2 jcm-12-00885-t002:** Effects over time: Primary outcomes ^1^.

Measurement (Range)	Control Group M (SD)	Intervention Group M (SD)	Average between-Group Difference (CI] ^1^	*p*-Value
Pain NRS ^a^ (0–10)				
Baseline	6.9 (1.5)	6.4 (1.5)		
6 months	6.3 (1.9)	5.8 (1.9)	−0.4 (−1.4, 0.5)	0.36
12 months	6.6 (1.7)	5.5 (1.9)	−0.9 (−1.9, 0.1)	0.08
Overall			−0.9 (−1.9, 0.05)	0.06
WPI ^b^ (1–21)				
Baseline	8.6 (5.0)	5.3 (3.6)		
6 months	8.0 (4.9)	4.8 (4.1)	−0.9 (−2.3, 0.6)	0.25
12 months	7.9 (4.8)	5.6 (4.9)	−1.8 (−3.5, −0.2)	0.03 *
Overall			−1.2 (−2.5, 0.1)	0.06
HR-QoL ^c^ (0–1)				
Baseline	0.58 (0.08)	0.60 (0.06)		
6 months	0.58 (0.06)	0.62 (0.07)	0.01 (−0.03, 0.04)	0.67
12 months	0.64 (0.07)	0.65 (0.07)	0.05 (0.02, 0.08)	<0.01 *
Overall			0.02 (−0.02, 0.05)	0.33

^1^ Adjusted for baseline characteristics and the propensity score. ^a^ Numeric Rating Scale; ^b^ Widespread Pain Index; and ^c^ Health-related Quality of Life. * Significant at the <0.05 level.

**Table 3 jcm-12-00885-t003:** Effects over time: Secondary outcomes.

Measurement (Range)	Control Group M (SD)	Intervention Group M (SD)	Average between-Group Difference [CI] ^1^	*p*-Value
RAND-36 Physical functioning (0–100)				
Baseline	48.0 (22.1)	61.0 (22.1)		
6 months	50.2 (19.5)	57.3 (24.7)	1.4 [−9.5, 12.4]	0.80
12 months	53.0 (23.1)	54.3 (27.5)	−3.0 [−14.6, 8.5]	0.61
Overall			−3.5 [−14.2, 7.1]	0.51
RAND-36 Role functioning: Physical (0–100)				
Baseline	26.3 (40.2)	19.1 (32.6)		
6 months	29.3 (38.4)	29.8 (43.0)	12.7 [−6.3, 31.7]	0.19
12 months	28.6 (36.5)	34.5 (40.7)	10.4 [−10.0, 30.8]	0.32
Overall			8.5 [−10.3, 27.3]	0.37
RAND-36 Energy/Fatigue (0–100)				
Baseline	36.1 (17.5)	46.3 (16.3)		
6 months	40.7 (15.5)	48.1 (20.2)	2.7 (−5.6, 10.9]	0.53
12 months	39.5 (18.5)	52.6 (19.7)	3.8 [−5.2, 12.8]	0.41
Overall			4.5 [−2.4, 11.3]	0.20
RAND-36 Emotional Wellbeing (0–100)				
Baseline	62.2 (21.1)	71.2 (16.1)		
6 months	63.3 (17.1)	66.8 (17.7)	2.3 [−4.8, 9.5]	0.52
12 months	63.6 (18.7)	69.5 (17.5)	5.0 [−2.7, 12.7]	0.21
Overall			5.6 [−0.5, 11.7]	0.07
RAND-36 Role functioning: Emotional (0–100)				
Baseline	51.8 (48.8)	66.7 (41.8)		
6 months	49.4 (48.5)	53.8 (49.1)	0.7 [−24.0, 25.5]	0.96
12 months	66.7 (45.4)	68.3 (44.1)	5.5 [−17.4, 28.4]	0.64
Overall			3.4 [−17.8, 24.6]	0.75
RAND-36 Social Functioning (0–100)				
Baseline	40.0 (22.2)	47.4 (22.1)		
6 months	45.2 (20.1)	47.3 (22.5)	0.3 [−7.9, 8.5]	0.95
12 months	45.7 (22.8)	45.7 (23.6)	0.9 [−7.9, 9.7)	0.85
Overall			0.4 [−7.2, 7.9]	0.93
RAND-36 Pain (0–100)				
Baseline	35.8 (21.1)	40.4 (17.2)		
6 months	42.7 (15.5)	47.6 (17.4)	6.2 [−1.3, 14.8]	0.11
12 months	42.3 (17.7)	47.0 (22.4)	5.2 [−2.9, 13.3]	0.21
Overall			3.9 [−3.4, 11.2]	0.30
RAND-36 General Health (0–100)				
Baseline	34.8 (18.4)	43.4 (13.3)		
6 months	32.9 (16.7)	46.0 (19.8)	8.7 [0.07, 17.4]	<0.05 *
12 months	37.7 (16.9)	40.7 (18.0)	1.2 [−7.9, 10.4]	0.79
Overall			4.6 [−3.0, 12.1]	0.24
RAND-36 Health change (0–100)				
Baseline	31.6 (25.8)	44.9 (29.4)		
6 months	44.8 (24.4)	59.6 (20.1)	11.8 [−1.6, 25.3]	0.09
12 months	44.6 (24.9)	58.3 (24.2)	12.7 [−1.8, 27.3]	0.09
Overall			14.2 [1.2, 27.1]	0.03 *
CSI ^a^ (1–100)				
Baseline	56.2 (16.2)	40.0 (16.9)		
6 months	53.5 (16.4)	44.2 (20.2)	−0.0 [−5.6, 5.5]	0.99
12 months	50.7 (15.3)	40.5 (19.4)	0.8 [−5.1, 6.7]	0.78
Overall			−0.2 [−5.5, 5.0]	0.93
PCS ^b^ (0–52)				
Baseline	33.2 (13.0)	29.6 (10.3)		
6 months	31.4 (13.1)	27.8 (10.5)	−3.7 [−7.8, 0.4]	0.08
12 months	29.3 (12.5)	24.9 (9.8)	−3.9 [8.3, 0.4]	0.08
Overall			−2.8 [−6.6, 1.1]	0.16

^1^ Adjusted for the baseline characteristics and the propensity score. ^a^ Central Sensitization Inventory; ^b^ Pain Catastrophizing Scale. * Significant at the <0.05 level.

**Table 4 jcm-12-00885-t004:** Patients’ satisfaction with treatment (CQ-Index Module Pain).

Item (Range)	Control Group M (SD)	Intervention Group M (SD)	Mean Difference [95% CI]	*p*-Value
Taken seriously (0–3)	1.9 (0.8)	2.3 (0.7)	−0.4 [−0.9, 0]	0.04 *
Trust in expertise (0–3)	1.9 (0.8)	2.1 (0.7)	−0.2 [−0.7, 0.2]	0.30
Rating of result (0–3)	1.6 (0.7)	2.0 (1.0)	−0.4 [−0.9, 0.1]	0.09
Rating of care (0–10)	6.5 (2.0)	6.7 (2.1)	−0.1 [−1.3, 1.1]	0.82

* Significant at the <0.05 level.

**Table 5 jcm-12-00885-t005:** Hoc Analysis on IPQ-B ^a^ and HADS ^b^.

Item (Range)	Control Group M (SD)	Intervention Group M (SD)	Average between-Group Difference [CI] ^1^	*p*-Value
IPQ-B 1: Consequences (0–10)				
Baseline	7.7 (1.9)	8.1 (1.8)		
6 months	7.8 (2.1)	6.9 (2.4)	−1.4 [−2.4, −0.4]	<0.01 *
12 months	7.9 (2.2)	6.9 (2.7)	−1.5 [−2.6, −0.5]	<0.01 *
Overall			−1.3 [−2.3, −0.3]	<0.01 *
IPQ-B 2: Timeline (0–10)				
Baseline	8.5 (2.1)	8.4 (2.0)		
6 months	9.3 (1.6)	7.9 (2.2)	−1.4 [−2.5, −0.3]	0.01 *
12 months	9.0 (1.8)	8.1 (2.7)	−1.5 [−2.5, −0.4]	<0.01 *
Overall			−1.4 [−2.6, −0.3]	0.01 *
IPQ-B 3: Personal Control (0–10)				
Baseline	4.5 (3.0)	5.3 (3.0)		
6 months	5.8 (2.7)	5.7 (2.8)	−0.6 [−2.0, 0.8]	0.41
12 months	5.1 (2.5)	5.9 (3.2)	0.2 [−1.3, 1.8]	0.78
Overall			0.3 [−0.9, 1.5]	0.65
IPQ-B 4: Treatment Control (0–10)				
Baseline	4.9 (3.0)	6.9 (1.7)		
6 months	6.8 (2.6)	6.6 (1.9)	−0.9 [−2.1, 0.4]	0.17
12 months	6.1 (2.2)	6.0 (3.0)	−1.1 [−2.2, 0.2]	0.10
Overall			−0.4 [−1.5, 0.8]	0.52
IPQ-B 5: Identity (0–10)				
Baseline	7.7 (2.0)	8.1 (1.4)		
6 months	7.6 (2.1)	7.1 (2.4)	−1.1 [−2.2, 0.0]	0.05
12 months	7.7 (2.2)	7.1 (2.6)	−1.3 [−2.5, −0.1]	<0.05 *
Overall			−1.0 [−2.2, 0.2]	0.10
IPQ-B 6: Concern (0–10)				
Baseline	5.0 (3.2)	7.0 (2.4)		
6 months	5.9 (3.0)	5.1 (3.0)	−3.0 [−4.2, −1.7]	<0.01 *
12 months	5.2 (3.2)	5.5 (2.8)	−2.1 [−3.4, −0.8]	<0.01 *
Overall			−2.2 [−3.4, −1.0]	<0.01 *
IPQ-B 7: Understanding (0–10)				
Baseline	6.2 (2.8)	5.4 (2.3)		
6 months	6.7 (2.6)	6.5 (2.7)	0.9 [−0.5, 2.3]	0.22
12 months	6.9 (2.8)	6.1 (3.1)	0.2 [−1.3, 1.7]	0.83
Overall			0.4 [−0.9, 1.6]	0.62
IPQ-B 8: Emotional Response (0–10)				
Baseline	7.1 (2.5)	7.4 (2.4)		
6 months	7.8 (2.4)	6.9 (3.0)	−1.5 [−2.6, −0.4]	<0.01 *
12 months	7.1 (2.6)	6.3 (2.8)	−1.1 [−2.3, 0.1]	0.07
Overall			−1.6 [−2.6, −0.5]	<0.01 *
HADS Anxiety (0–21)				
Baseline	12.1 (2.8)	13.0 (1.9)		
6 months	11.9 (2.4)	12.9 (2.1)	0.1 [−0.8, 1.1]	0.78
12 months	12.1 (2.6)	12.3 (2.0)	0.6 [−0.3, 1.6]	0.17
Overall			0.4 [−0.6, 1.3]	0.45
HADS Depression (0–21)				
Baseline	8.1 (1.9)	8.8 (1.6)		
6 months	8.8 (1.9)	9.2 (1.4)	0.8 [−0.2, 1.8]	0.11
12 months	8.9 (1.9)	10.0 (1.4)	0.1 [−0.8, 1.1]	0.76
Overall			0.2 [−0.6, 1.1]	0.58

^1^ Adjusted for the baseline characteristics and the propensity score. ^a^ Brief Illness Perception Questionnaire; ^b^ Hospital Anxiety and Depression Scale. * Significant at the <0.05 level.

## Data Availability

The data presented in this study are available on request from the corresponding author. The data are not publicly available due to privacy reasons.

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
