# Peer review of "Effectiveness of a Primary Care Multidisciplinary Treatment for Patients with Chronic Pain Compared with Treatment as Usual"

_jcm, 2023, doi:10.3390/jcm12030885_

Round 1
Reviewer 1 Report
Thank you for a very interesting and well-performed study. I have only minor issues, which could be clarified in the text.
1. The study is non-randomized. Intervention group were recruited from 3 clinics. Could you explain more enrollment process? Consecutive patients. Were controls enrolled and treated at the same clinics?
2. Could you describe more comprehensively the nature of TAU?
3. The study had a high drop-out rate. I have understood that your statistical method did not include imputation of missing values. Could you explain more how you considered drop-out rate?
4. Maybe include the high drop-out rate as a limitation?
5. Was the competence of the intervention team checked after training? After all, 2 + 2 day training is very short!
6. On page 4 (in the beginnning) you explain that .."biopsychosocial model of the patient, their pain..". If you did had individualized treatment, shouldn't it be .."his/her pain"?
7. Patient evaluation took 3 hours. How long did physical examination on average take?
8. In discussion (page 13) you speculated that the negative results may be because of too short pain neuroscience education training? However, Traeger et al 2019 did evaluate the duration of the education and found that a shorter education equals a longer one. Your thoughts about this?
9. In discussion, you could also consider discussing CFT which is a different approach to chronic pain.
10. You have not at all discussed the severity of symptoms of your patients (considering that they had e.g. pain sensitization). Not easy patients to treat. Probably they should be evaluated and treated much earlier. You could add something of this too (although you have mentioned it in discussion)?
Author Response
We are happy with your positive feedback and have completed the revision. We have addressed all issues raised and feel that this has substantially improved the quality of the work. Therefore, we want to thank you for the helpful comments and guidance on improving our work. All revisions are explained below and highlighted in the text using tracked changes.
- The study is non-randomized. Intervention group were recruited from 3 clinics. Could you explain more enrollment process? Consecutive patients. Were controls enrolled and treated at the same clinics?
Response: A more detailed description of the enrolment process has been added in the section between lines 109-124: “General practitioners (GPs) working at these practices identified eligible patients in their practice, and invited them to take part in the multidisciplinary program. The multi/transdisciplinary program under study was implemented in these practices several years before the trial started. All patients who entered the multidisciplinary program in these practices and met the in- and exclusion criteria were asked to participate in the trial. All patients were provided with an informed consent form, which they were asked to sign upon entering the trial. Participants for the control group were recruited via advertisements at social media (i.e., Facebook). Control group patients did not receive the treatment under study, receiving treatment as usual instead, at different general practices as the general practices providing the intervention. To keep both groups as similar as possible, only patients living in the same geographical area as the patients in the intervention group (i.e. the North of the Netherlands) were recruited for the control group.”
Could you describe more comprehensively the nature of TAU?
Response: TAU is now described more thoroughly between lines 215-226:
“The key elements of the guideline are as follows:
- Education about chronic pain and the biopsychosocial model
- Non-drug treatment: advise to seek distraction and support and to find a good balance between relaxation and activity. Possible referral to a psychologist in case of harmful cognitions, emotions and behaviors, possible referral to a physiotherapist or remedial therapist for exercise programs aimed at an active lifestyle, possible referral to social worker if social problems play a role.
- Drug treatment: avoid drug treatment as possible, especially avoid opioids if possible. When prescribing drugs, aim to prescribe for short period.
- Referrals: Referral to a rehabilitation if patient is experiencing a high degree of limitations due to pain, referral to medical specialist is there is a possible underlying cause that is treatable, etc.”
- The study had a high drop-out rate. I have understood that your statistical method did not include imputation of missing values. Could you explain more how you considered drop-out rate?
Response: Thank you for alerting us to this important issue. As we performed a mixed-model analysis on longitudinal data, it was not deemed necessary to use multiple imputation of missing values (following the study by Twisk et al. 2013). Twisk and colleagues argue that a mixed model analysis is able to handle missing values well through its use of maximum likelihood estimation and results in a valid estimates. To account for your issue, we have now added this to the body of the paper where it is now discussed in lines 518-525: “Second, the sample size was relatively small, and the dropout of participants was considerable (i.e. 67.4% in the intervention group and 56.5% in the control group). However, the sample size was sufficient and as Twisk et al. [52] argued, longitudinal mixed models are well able to deal with missing values through their use of maximum likelihood estimation and result in valid estimates [53]. In addition, there was no differential dropout or attrition, as in that drop-out rates were similar in both arms (control group and intervention group) and therefore, the risk of biased result is considered low.”
- Maybe include the high drop-out rate as a limitation?
Response: Thank you for alerting us to this issue. We agree with the reviewer and the high drop-out rate is already described as a limitation on lines 518-525, additionally, we have added the specific drop-out rates: “Second, the sample size was relatively small, and the dropout of participants was considerable (i.e. 67.4% in the intervention group and 56.5% in the control group). However, the sample size was sufficient and as Twisk et al. [52] argued, longitudinal mixed models are well able to deal with missing values through their use of maximum likelihood estimation and result in valid estimates [53]. In addition, there was no differential dropout or attrition, as in that drop-out rates were similar in both arms (control group and intervention group), and therefore, the risk of biased result is considered low.”
- Was the competence of the intervention team checked after training? After all, 2 + 2 day training is very short!
Response: No, the reviewer is correct in that it was not checked and this is described as a limitation in 440-449. In addition, a line was added with a recommendation for further studies: “In addition, it may be worthwhile investigating the effect of offering more or longer pain neuroscience education sessions, as a recent systematic review found favorable effects of longer duration of PNE, although this needs further study [47]. Furthermore, the primary health care providers in the intervention were trained to work with the protocol. Some of them were however biomedically trained doctors and physiotherapists. It is a possibility that the total amount of training for the primary health care providers was too short to get acquainted with the protocol. Further studies may investigate the effect of more extensive training of the multidisciplinary team providing the intervention, and in addition, check whether the team’s attitude and behavior towards chronic pain did indeed change after training.”
In addition, a paragraph was added on lines 527-532, which discusses the lack of a fidelity check and describes recommendations for further research: “Likewise, while the healthcare professionals in the intervention group received training and were instructed to provide treatment according to protocol, it was not checked whether they actually adhered to protocol. Future research is therefore recommended to include a fidelity check of the intervention.”
- On page 4 (in the beginning) you explain that .."biopsychosocial model of the patient, their pain..". If you did had individualized treatment, shouldn't it be .."his/her pain"?
Response: I believe that ‘their pain’ is also correct, as singular ‘they/them/their’ can be used when referring to a generic person whose gender is unknown or irrelevant (i.e. the patient, whose gender we do not know and is irrelevant in this case). See also: https://apastyle.apa.org/blog/singular-they Still, if you prefer us to revise it according to your suggestions, please let us know and we are happy to do so.
- Patient evaluation took 3 hours. How long did physical examination on average take?
Response: This varied between patients, but on average, about half of the GP’s assessment and half of the physiotherapist assessment was spent on physical examination, resulting in 60 minutes in total. A line has been added to specify this more clearly (line 160-162): “Both the GP and the physiotherapist conducted a physical examination (both ±30 minutes in duration).”
- In discussion (page 13) you speculated that the negative results may be because of too short pain neuroscience education training? However, Traeger et al 2019 did evaluate the duration of the education and found that a shorter education equals a longer one. Your thoughts about this?
Response: In the discussion, we meant that the training of the healthcare professionals in order to provide PNE may have been too short, not the PNE session itself. As this might not have been clear enough in the previous version of the manuscript, we have discussed this more clearly on line 443-449: “Furthermore, the primary health care providers in the intervention were trained to work with the protocol. Some of them were however biomedically trained doctors and physiotherapists. It is a possibility that the total amount of training for the primary health care providers was too short to get acquainted with the protocol. Further studies may investigate the effect of more extensive training of the multidisciplinary team providing the intervention, and in addition, check whether the team’s attitude and behavior towards chronic pain did indeed change after training.”
Responding to your remark about the research by Traeger et al: The study by Traeger et al. (2019) on intensive patient education was performed in patients with acute low back pain, thus patients had specific pain for fewer than 6 weeks. The patients in this study had non-specific chronic pain, thus often generalized pain for more than 6 months. We believe a comparison is difficult between these studies, as psychosocial factors are more likely to play a (more complex) role in chronic pain than in acute pain. Furthermore, besides not investigating patients with chronic pain, Traeger et al. did not compare intensive patient education with a shorter patient education, but in fact compared intensive patient education with a placebo intervention, which did not include an education component at all. A recent systematic review by Watson et al. (2019) did, however, find favourable effects on longer duration of pain neuroscience education on patients with chronic musculoskeletal pain and recommended further research. A line discussing this study has been added on lines 440-443: “In addition, it may be worthwhile investigating the effect of offering more or longer pain neuroscience education sessions, as a recent systematic review found favorable effects of longer duration of PNE, although this needs further study [47].”
- In discussion, you could also consider discussing CFT which is a different approach to chronic pain.
Response: A recent systematic review (Miki et al. 2022) on the effectiveness of CFT concluded that there was very little evidence that CFT was more effective than other interventions for reducing disability for chronic nonspecific low back pain. We therefore believe that it is safer to stick to more well-studied interventions such as CBT, until CFT’s effectiveness is better researched in larger RCT’s and there is more evidence for its effectiveness. Therefore, CFT is not recommended in this study.
See also: Miki T, Kondo Y, Kurakata H, Buzasi E, Takebayashi T, Takasaki H. The effect of cognitive functional therapy for chronic nonspecific low back pain: a systematic review and meta-analysis. Biopsychosoc Med. 2022 May 21;16(1):12. doi: 10.1186/s13030-022-00241-6. PMID: 35597961; PMCID: PMC9123771.
- You have not at all discussed the severity of symptoms of your patients (considering that they had e.g. pain sensitization). Not easy patients to treat. Probably they should be evaluated and treated much earlier. You could add something of this too (although you have mentioned it in discussion)?
Response: We have indeed mentioned this in lines 428-435: “In the current study, both the participants in the intervention group and in the control group had had their pain for a substantial amount of time prior to the intervention (103 months in the control group and 93 months in the intervention group). Chronicity of pain has been shown to be associated with only modest changes in pain severity, self-reported disability, pain catastrophizing, and fear of movement/re-injury [45]. This seems to underline the need for an earlier multidisciplinary approach in patients with chronic pain, as not recognizing psychosocial risk factors can lead to chronification of pain [46].”
Reviewer 2 Report
This study evaluated the effectiveness of multidisciplinary treatment for chronic pain embedded in primary care (n= 43), compared to treatment as usual (n=46). The results show that there were no statistically significant effects.
I am sure that a lot of work was invested in this study and the results can be frustrating.
Some questions/comments:
1. Why did you choose a non-randomized control trial?
2. The allocation of the participants across study groups was not randomized and caused a selection bias – especially - most participants in the control group did not have paid employment, while the majority of the participants in the intervention group were employed.
3. As you stated in the limitation paragraph, the sample size was small
4. Did you check the fidelity of the intervention?
Author Response
We are happy with your positive feedback and have completed the revision. We have addressed all issues raised and feel that this has substantially improved the quality of the work. Therefore, we want to thank you for the helpful comments and guidance on improving our work. All revisions are explained below and highlighted in the text using tracked changes.
Some questions/comments:
- Why did you choose a non-randomized control trial?
Response: Unfortunately, a randomized control trial was not possible, because of the nature of the intervention and the lack of resources to randomize entire practices. Patients in the intervention group remained within their own general practice in primary care and their GP invited them to enter in the multidisciplinary program if they were eligible. Patients in the control group were treated at different general practices, where treatment as usual was provided. Thus: all patients remained at the general practices that they were already registered at, making it impossible to randomize patients.
As this limitation may have not been described clearly enough, we have more clearly addressed this issue on lines 512-518: “First, the allocation of the participants across study groups was not randomized. Randomization was not possible at patient or cluster level, because of the nature of the intervention and the lack of resources to randomize entire practices. Patients in both the intervention group and control group remained at the general practices that they were already registered at, making it impossible to perform an RCT or CRT. Therefore, convenience sampling was used. This could have caused a selection bias, which we tried to reduce by using propensity score weighting.”
- The allocation of the participants across study groups was not randomized and caused a selection bias – especially - most participants in the control group did not have paid employment, while the majority of the participants in the intervention group were employed.
Response: The non-randomized design indeed has the potential for selection bias. This was managed by using propensity score weighting, in which both groups were weighted to make them as comparable as possible, essentially mimicking a randomized controlled trial. With propensity scores, the individual’s probability of being assigned to the intervention or control group given baseline characteristics is calculated. This is described on lines 297-303: “To deal with the non-randomized nature of this study, effect differences were adjusted for patients’ propensity score. The propensity score indicates the probability of a patient being assigned to the intervention group, given a set of baseline characteristics. The propensity score was estimated based on the following possible confounding variables: whether patients had other complaints besides pain, whether patients had their pain symptoms for the first time or not, the duration of their pain at baseline, employment status, education level, age, and sex using the pscore package in STATA [43].”
- As you stated in the limitation paragraph, the sample size was small
Response: The reviewer is correct. We have therefore acknowledged this limitation in line 518-525: “Second, the sample size was relatively small, and the dropout of participants was con-siderable (i.e. 67.4% in the intervention group and 56.5% in the control group). However, the sample size was sufficient and as Twisk et al. [52] argued, longitudinal mixed models are well able to deal with missing values through their use of maximum likeli-hood estimation and result in valid estimates [53]. In addition, there was no differential dropout or attrition, as in that drop-out rates were similar in both arms (control group and intervention group), and therefore, the risk of biased result is considered low.”
- Did you check the fidelity of the intervention?
Response: The reviewer is correct that we did not check the fidelity of the intervention. This has therefore been acknowledged as a limitation on lines 525-532. A line has also been added recommending checking fidelity in future research: “Third, it was not checked whether the participants in the control group were actually treated according to the guideline for managing patients with chronic pain; the “NHG Standaard Pijn” [14]. Likewise, while the healthcare professionals in the intervention group received training and were instructed to provide treatment according to protocol, it was not checked whether they actually adhered to protocol. Future research is therefore recommended to include a fidelity check of the intervention. Furthermore, it is recommended to investigate dose-response relationships within the intervention.”
Reviewer 3 Report
The authors have presented an interesting nonrandomized trial of a multidisciplinary pain treatment in primary care. Overall this is a well described intervention study but the manuscript could be strengthened by better clarity in some of the methods and why certain analyses that might have been conducted were not presented. First, a major issue is the potential nonequivalence of the control group as well as unknown level of treatment. While the authors indicate propensity adjustments were made to account for potential baseline differences it would be helpful to see actual statistical comparisons between the two groups at baseline instead of just descriptive values presented in Table 1. Also, what characteristics were used to create the propensity scores.? In addition, more information about the Intent to Treat analysis would be helpful given the large level of attrition in both groups, but especially in the intervention group., resulting in final sample sizes well below the described power calculations.
The results presented in Tables 2, & 5 as 'average between group differences' are also not well explained. Presumably baseline scores and other factors were used to calculate these scores?
It is also unclear why analyses Illness Perceptions and Anxiety and Depression are labelled as 'post-hoc". Further, in the discussion there is a point made about the intervention group having more depression possibly explaining the largely nonsignificant findings. This should be explored analytically.
Finally, it appears it's possible to investigate dose-response relationships within the intervention group, as well as adjust for baseline characteristics and scores. Such analyses might help tease out better who responded to this intervention. This is especially important since we know so little about the treatment received by the control group.
Minor notes: lines 23-24 in abstract are redundant. Lines 349-350 reporting the Rand scale are unclear-- what is meant by statistically significant findings but 'overall effect was not statistically significant'??
Author Response
We are happy with your positive feedback and have completed the revision. We have addressed all issues raised and feel that this has substantially improved the quality of the work. Therefore, we want to thank you for the helpful comments and guidance on improving our work. All revisions are explained below and highlighted in the text using tracked changes.
- First, a major issue is the potential nonequivalence of the control group as well as unknown level of treatment. While the authors indicate propensity adjustments were made to account for potential baseline differences it would be helpful to see actual statistical comparisons between the two groups at baseline instead of just descriptive values presented in Table 1.
Response: Although we understand your concern, we respectfully disagree with this comment. As de Boer et al. (2015) argued: According to the CONSORT statement, significance testing of baseline differences in randomized controlled trials should not be performed. In fact, this practice has been discouraged by numerous authors throughout the last forty years. Testing for baseline differences is deemed purposeless and can even be misleading, especially when using these tests as a basis for choosing covariates for analysis. It ignores the strength of covariates as prognostic variates. In other words: there can be differences between groups (e.g. as a result of small groups) which are not statistically significant, while these difference can however have prognostic value. Vice versa, statistically significant differences between groups do not necessarily mean that these differences have prognostic value. Therefore, we prefer not to present these p-values in this article.
See also: de Boer, Michiel R et al. “Testing for baseline differences in randomized controlled trials: an unhealthy research behavior that is hard to eradicate.” The international journal of behavioral nutrition and physical activity vol. 12 4. 24 Jan. 2015, doi:10.1186/s12966-015-0162-z
- Also, what characteristics were used to create the propensity scores.?
Response: This is described more clearly in lines 300-303: “The propensity score was estimated based on the following possible confounding variables: whether patients had other complaints besides pain, whether patients had their pain symptoms for the first time or not, the duration of their pain at baseline, employment status, education level, age, and sex using the pscore package in STATA [43].”
- In addition, more information about the Intent to Treat analysis would be helpful given the large level of attrition in both groups, but especially in the intervention group., resulting in final sample sizes well below the described power calculations.
Response: As we performed a mixed-model analysis on longitudinal data, all cases included at baseline were analysed, resulting in sufficient final sample sizes. Twisk and colleagues argue that a mixed model analysis is able to handle missing values well through its use of maximum likelihood estimation and results in a valid estimates. This is now discussed in lines 518-525: “Second, the sample size was relatively small, and the dropout of participants was considerable (i.e. 67.4% in the intervention group and 56.5% in the control group). However, the sample size was sufficient and as Twisk et al. [52] argued, longitudinal mixed models are well able to deal with missing values through their use of maximum likelihood estimation and result in valid estimates [53]. In addition, there was no differential dropout or attrition, as in that drop-out rates were similar in both arms (control group and intervention group), and therefore, the risk of biased result is considered low.”
- The results presented in Tables 2, & 5 as 'average between group differences' are also not well explained. Presumably baseline scores and other factors were used to calculate these scores?
Response: The estimates were corrected by baseline characteristics and the propensity scores. This is now added as a footnote under table 2 and 4.In addition, this has been more clearly described in the body of the text on lines 347-352: “The values presented under average between group differences are model estimates of linear mixed-effect models and are corrected by baseline characteristics and propensity scores. The regression-coefficients can be interpreted as average differences between the intervention group and the control group at a certain follow-up point (6 or 12 months) compared with baseline. The overall effect measures can be interpreted as the effect over the total follow-up time of 12 months, instead of time x treatment effects.”
- It is also unclear why analyses Illness Perceptions and Anxiety and Depression are labelled as 'post-hoc".
Response: The analyses on the IPQ-B and the HADS were labelled as a post-hoc analysis because they were not described in the original study designed as registered in the trial register (https://trialsearch.who.int/Trial2.aspx?TrialID=NTR6014). This explanation has been added on lines 268-269: “Two outcome measures, which were not described in the original study protocol as registered in the trial register, were added in a post-hoc analysis.”
- Further, in the discussion there is a point made about the intervention group having more depression possibly explaining the largely nonsignificant findings. This should be explored analytically.
Response: Differences between levels of depression were actually not statistically significant. However, while between-group differences were not statistically significant, it possible that (differences in) levels of depression do have prognostic value (see our previous comment about baseline differences). Therefore, even when not considering statistically significant differences in levels of depression, it is an important factor to consider. We feel that this is an important issue to address, therefore we have added further explanation of this on lines 457– 463: “Lastly, patients in both the control group and the intervention group reported significant levels of depression (HADS >8) at baseline, and symptoms of depression appeared to increase over time. Research has shown that depression is a predictor of poor outcome for multidisciplinary treatment and quality of life in patients with chronic pain [49,50]. While levels of depression were equal in both groups (and differences were not statistically significant), this could dampen the effect of treatment success.”
- Finally, it appears it's possible to investigate dose-response relationships within the intervention group, as well as adjust for baseline characteristics and scores. Such analyses might help tease out better who responded to this intervention. This is especially important since we know so little about the treatment received by the control group.
Response: Regarding your comment to adjust for baseline characteristics and scores, we have done this and described this in our methods between lines 297-303 “To deal with the non-randomized nature of this study, effect differences were adjusted for patients’ propensity score. The propensity score indicates the probability of a patient being assigned to the intervention group, given a set of baseline characteristics. The propensity score was estimated based on the following possible confounding variables: whether patients had other complaints besides pain, whether patients had their pain symptoms for the first time or not, the duration of their pain at baseline, employment status, education level, age, and sex using the pscore package in STATA [43].” and as a footnote under tables 2,3 and 5. Unfortunately, we do not have any information about treatment fidelity or dose-response, and can therefore not perform a per-protocol analysis. We have added the lack of a treatment fidelity-check as a limitation and have added a recommendation for future research to look in to dose-response relationships on lines 525-532: “Third, it was not checked whether the participants in the control group were actually treated according to the guideline for managing patients with chronic pain; the “NHG Standaard Pijn” [14]. Likewise, while the healthcare professionals in the intervention group received training and were instructed to provide treatment according to protocol, it was not checked whether they actually adhered to protocol. Future research is therefore recommended to include a fidelity check of the intervention. Furthermore, it is recommended to investigate dose-response relationships within the intervention.”
Minor notes: lines 23-24 in abstract are redundant.
Response: we have changed the line in: “The intervention consisted of pain neuroscience education and treatment by a GP, psychologist and physiotherapist.”
Lines 349-350 reporting the Rand scale are unclear-- what is meant by statistically significant findings but 'overall effect was not statistically significant'??
Response: By this is meant that there was a statistically significant between-group difference between baseline and 6 months, but the corresponding overall effect (meaning the effect over the whole 12 months) was not statistically significant. We have described this more clearly on lines 373-376: “At 6 months, the intervention group rated their overall health (as measured on the RAND-36 General Health-item) statistically significantly better than the control group, but the corresponding overall effect (over the whole 12 months) was not statistically significant.” In addition, we hope that the previously mentioned explanation on lines 347-352 clear this up: “The values presented under average between group differences are model estimates of linear mixed-effect models and are corrected by baseline characteristics and propensity scores. The regression-coefficients can be interpreted as average differences between the intervention group and the control group at a certain follow-up point (6 or 12 months) compared with baseline. The overall effect measures can be interpreted as the effect over the total follow-up time of 12 months, instead of time x treatment effects.”
Round 2
Reviewer 3 Report
The authors have done a good job improving the description of the analytical methods. and addressing prior review comments. One important concern remains, and that is the lack of knowledge about what type/how much treatment each group received and length of treatment. I don't believe we can state this as a fair evaluation of multidisciplinary pain management in primary care without knowing what the clinicians actually implemented after training. Also, given some of the 6-month differences in the intervention, and overall clinically significant levels of reported change of various variables in the intervention group, some clearer mention should be made that the intervention might not have been active for more than a few months and therefore the 12-month endpoint is quite a bit past end of active treatment. This would be consistent with the authors mentioning intensity and booster sessions but an additional sentence or two would be useful. Also not taken into account is how much completion of the self-assessments by all patients, but specifically controls might change their reporting of outcomes over time. While the authors have addressed the lack of documentation of intensity of the intervention as a limitation, in future recommendations, it should be made more prominent as a discussion point and in study limitations. As the manuscript reads in many places now it appears the authors are more definitively stating multidisciplinary pain management in primary care is not valuable, whereas I read some of the data and the limitations of the study as being more equivocal on the conclusions.
Author Response
Response: Thank you so much for your positive feedback and the opportunity to further improve our manuscript. We want to thank you for the helpful comments and guidance on improving our work. All revisions are explained below and highlighted in the text using tracked changes.
- One important concern remains, and that is the lack of knowledge about what type/how much treatment each group received and length of treatment. I don't believe we can state this as a fair evaluation of multidisciplinary pain management in primary care without knowing what the clinicians actually implemented after training. Also, given some of the 6-month differences in the intervention, and overall clinically significant levels of reported change of various variables in the intervention group, some clearer mention should be made that the intervention might not have been active for more than a few months and therefore the 12-month endpoint is quite a bit past end of active treatment. This would be consistent with the authors mentioning intensity and booster sessions but an additional sentence or two would be useful.
Response: Thank you for pointing out this issue. We agree that this should be further clarified in the body of the text. We have added the following paragraph on lines 458-466: “In addition, at 12 months, the PNE-session had been 12 months in the past for patients, and the intervention (i.e. sessions with the physiotherapist and/or the psychologist) may not have been active for several months for many patients at this stage as well. Looking at the results, most of the changes in the intervention group as well as between-group differences appeared to happen in the first 6 months, when the patients were often still in active therapy. Further studies could focus on adapting treatment, such as adding booster sessions in which the pain neuroscience education is repeated after a certain amount of time, or offer more intensive treatment in an effort to improve the long-term effects of the intervention”.
- Also not taken into account is how much completion of the self-assessments by all patients, but specifically controls, might change their reporting of outcomes over time.
Response: The reviewer is correct in that filling out the self-assessments itself may have influence the patients’ reporting over time. However, we believe that the effect of self-reporting would be equal in both groups. We have added a paragraph addressing this on lines 504-511: “It is possible that reporting on pain and pain-related outcomes itself had an effect on patients’ outcomes in both the intervention group and the control group. Research has shown that attention to pain and mood both alter pain perception, and filling out the self-reports may have drawn attention to the pain and pain-related outcomes and hence influenced the patients’ experience of their pain and its outcomes [51]. However, as both the control group and the intervention group received treatment (treatment as usual vs. intervention,) the effect of self-reporting is believed to be equal in both groups and it is not deemed to have led to bias.”
- While the authors have addressed the lack of documentation of intensity of the intervention as a limitation, in future recommendations, it should be made more prominent as a discussion point and in study limitations.
Response: Thank you for pointing out this issue. We believe the reviewer is correct in stating that this should be made more prominent as a limitation. A paragraph has been added in the discussion to address this on line 569-577: “The intensity of the intervention (i.e. the amount of therapy sessions a patient received from the physiotherapist and/or psychologist, the exact nature of the therapy sessions, how long the total duration of the intervention was for every patient, etc.) was not documented either and the effect of intervention-intensity (i.e. dose-response relationship) could therefore not be taken into account. Future research is therefore recommended to include a fidelity check of the intervention. Furthermore, for future studies it is recommended to carefully document the type and frequency of therapy sessions within the intervention, to be able to investigate dose-response relationships.”
- As the manuscript reads in many places now it appears the authors are more definitively stating multidisciplinary pain management in primary care is not valuable, whereas I read some of the data and the limitations of the study as being more equivocal on the conclusions.
Response: We believe that the reviewer is right in stating that the conclusion that multidisciplinary treatment in primary care is not valuable is possibly too rash. We have therefore added a line adding nuance to the conclusion on line 588-596: “Based on these results, the findings do not support implementation of a low intensity outpatient multidisciplinary primary care treatment to treat chronic pain. However, as a result of several study limitations, it is considered unwarranted to conclude that multidisciplinary treatment in primary care is not valuable at all. Future studies may aim to address the listed limitations to aim for a fairer, more detailed comparison between the intervention and TAU. Multidisciplinary treatment in primary care for chronic pain may very well be effective as well as cost-effective, but further research is necessary to investigate how the intervention has to be set up in order for it to be significantly more effective than TAU in managing chronic pain.”
We have also changed the conclusion in the abstract in line with this, as follows: “Based on the results, the findings do not support effectiveness of a low intensity outpatient multidisciplinary primary care treatment to treat chronic pain compared to TAU. However, as a result of several study limitations, it is considered unwarranted to conclude that multidisciplinary treatment in primary care is not valuable at all.”